# Effect of Chemically Induced Hypoxia on Osteogenic and Angiogenic Differentiation of Bone Marrow Mesenchymal Stem Cells and Human Umbilical Vein Endothelial Cells in Direct Coculture

**DOI:** 10.3390/cells9030757

**Published:** 2020-03-19

**Authors:** Van Thi Nguyen, Barbara Canciani, Federica Cirillo, Luigi Anastasia, Giuseppe M. Peretti, Laura Mangiavini

**Affiliations:** 1IRCCS Istituto Ortopedico Galeazzi, 20161 Milan, Italy; thivan.nguyen@grupposandonato.it (V.T.N.); barbara.canciani@grupposandonato.it (B.C.); giuseppe.peretti@unimi.it (G.M.P.); 2Stem Cells for Tissue Engineering Laboratory, IRCCS Policlinico San Donato, 20097 San Donato Milanese, Italy; federica.cirillo@grupposandonato.it (F.C.);; 3Università Vita-Salute San Raffaele, 20132 Milan, Italy; 4Department of Biomedical Sciences for Health, University of Milan, 20133 Milan, Italy

**Keywords:** BMSC, HUVEC, normoxia, hypoxia, osteogenesis, angiogenesis

## Abstract

Bone is an active tissue where bone mineralization and resorption occur simultaneously. In the case of fracture, there are numerous factors required to facilitate bone healing including precursor cells and blood vessels. To evaluate the interaction between bone marrow-derived mesenchymal stem cells (BMSC)—the precursor cells able to differentiate into bone-forming cells and human umbilical vein endothelial cells (HUVEC)—a cell source widely used for the study of blood vessels. We performed direct coculture of BMSC and HUVEC in normoxia and chemically induced hypoxia using Cobalt(II) chloride and Dimethyloxaloylglycine and in the condition where oxygen level was maintained at 1% as well. Cell proliferation was analyzed by crystal violet staining. Osteogenesis was examined by Alizarin Red and Collagen type I staining. Expression of angiogenic factor-vascular endothelial growth factor (VEGF) and endothelial marker-von Willebrand factor (VWF) were demonstrated by immunohistochemistry and enzyme-linked immunosorbent assay. The quantitative polymerase chain reaction was also used to evaluate gene expression. The results showed that coculture in normoxia could retain both osteogenic differentiation and endothelial markers while hypoxic condition limits cell proliferation and osteogenesis but favors the angiogenic function even after 1 of day treatment.

## 1. Introduction

Bone formation includes two important processes, namely intramembranous ossification that results in flat bones (bones of the face, most bones of cranium and clavicles) and endochondral ossification that forms long bones (tibia, femur, humerus, and radius) [1]. While the intramembranous process is characterized by initial differentiation of mesenchymal cells into osteoblasts that release osteoid (matrix composed of collagen and proteins) and then become osteocytes followed by blood vessel formation, endochondral pathway starts with the replacement of hyaline cartilage composed of chondroblasts differentiated from mesenchymal cells by perichondrium derived osteoblasts that secret bone matrix; in turn, this matrix prevents from the diffusion of the blood providing nutrients to the cartilage tissue leading to chondroblast apoptosis and cartilage tissue destruction, which cause the penetration of surrounding blood vessel into free spaces to create medullary cavity [2]. In both processes, contribution and the close link between mesenchymal cells and blood vessels are crucial. In a hip replacement procedure where damaged femoral head and acetabulum are replaced by a metal hip inserted to the femur, bone grafts are sometimes required to improve osteointegration between the implant and the host’s bone. These bone grafts when pressed into the space between the implant and the femur, they may also attract adjacent osteoblasts, mesenchymal stem cells to migrate and reside in the graft. Furthermore, the existing blood vessels present in bone marrow carry oxygen and nutrients to the cells and possibly promote angiogenesis in the bone graft. Despite high vasculature, bone marrow has been considered as a hypoxic tissue [3,4], thus the study of osteogenesis and angiogenesis in the absence of oxygen could be useful to investigate the physiological state and may be utilized to create an in vitro pre-vascularized engineered bone graft for future implantation.

In the literature, HUVEC was found to inhibit BMSC differentiation into osteoblast in indirect coculture (at 1:1 ratio) [5]. On the contrary, in the 1:1 direct coculture of HUVEC and the cells with MSC characteristics, such as dental pulp stem cells and dental follicle derived stem cells, both osteogenic and angiogenic potentials are increased [6,7] respectively. According to Yamamoto et al., 2019 [8] and Zhang et al. 2017 [9], direct coculture at the above ratio of HUVEC and BMSC also promotes angiogenic markers. We, therefore hypothesize that direct coculture of HUVEC and BMSC (1:1 ratio) may promote the generation of a pre-vascularized bone graft.

Hypoxia-inducible factor 1 (HIF-1) activates the expression of genes related to angiogenesis (VEGF), erythropoiesis (EPO), and metabolism (ALDA, ENO1, LDHA, PFKL, PGK1) [10]. However, Gawlitta et al. 2012 [11] demonstrated that coculture of BMSC with the endothelial colony-forming unit as pellets in the osteogenic medium in hypoxic conditions impedes vasculogenesis of engineered bone. It is consistent with the findings of Griffith et al., 2009 [12] where 1% oxygen hypoxia was proven to decrease in vitro angiogenesis of the engineered tissue composed of HUVEC and normal human lung fibroblasts—a type of stromal cells. It should be noted that a hypoxic incubator cannot guarantee complete hypoxia as once the incubator’s door is open, the cells inside could be in contact with oxygen and it also requires time for the incubator to lower the oxygen back to the required level. Although the hypoxic chamber overcomes this drawback as the working bench and trays for placing cell plates are inside the hypoxic chamber, it can be limited solely for the pilot experiment. To address these problems, we used two reagents previously known for simulating hypoxia namely Cobalt(II) chloride (CoCl_2_) and Dimethyloxaloylglycine (DMOG) for coculture system to compare with in normoxia.

In this study, we aimed to evaluate cell proliferation of BMSC and HUVEC in direct coculture in normoxia and hypoxia-induced by hypoxic agents. To better simulate the inflammatory condition, IL-1β was also added to the culture medium in normoxia. Furthermore, in vitro osteogenesis and angiogenesis were analyzed to determine the role of oxygen in these two processes. To our knowledge, this is the first study addressing the differentiation of BMSC and HUVEC in direct coculture system under the effect of hypoxia stimulated by chemical reagents. The findings of this study could be applied in the production of engineered bone grafts with osteoinductivity and pro-angiogenic characteristics for implantation.

## 2. Materials and Methods

### 2.1. Cell Isolation and Culture

#### 2.1.1. Isolation and Expansion of BMSC

Approval of the procedure was obtained from the Ethical Committee of IRCCS Istituto Ortopedico Galeazzi (Protocol v02 del 09/06/2017). Four patients undergoing hip replacement procedures were selected for the study with their informed consent. The patients aged between 50 and 80 years old with a body mass index (BMI) from 18 to 30 kg/m^2^. None of them had a history of disease related to infection with HIV or HBV/HCV, bone metabolism, and joint inflammation.

Bone marrow aspirate from the femoral canal was collected and washed twice with 1× Phosphate buffer saline (PBS) at 1,700× *g* for 10 min at room temperature (RT). Nucleated cells were counted and plated at density 5 × 10^4^ cells/cm^2^ in medium containing alpha minimum essential medium (αMEM, Thermofisher, Bleiswijk, Netherlands), 10% volume/volume (*v/v*) fetal bovine serum (FBS, Euroclone, Milan, Italy), and 1% (*v/v*) penicillin/streptomycin/glutamine (P/G/S, Thermofisher, Bleiswijk, Netherlands) supplemented with 1 ng/mL recombinant human FGF-2 (R&D Systems, Minneapolis, MN, USA). The primary cells were incubated in the 37 °C and 5% CO_2_ incubator for 1 week then the medium was changed twice per week. The cells were expanded up to Passage 4 (P4). We previously demonstrated that BMSC isolated following this protocol have typical characteristics of MSC such as they are devoid of hematopoietic stem cells markers (CD34, CD45) but express markers of MSC including CD73, CD90, and CD106; they are also able to differentiate into osteoblasts, adipocytes, and chondrocytes [13].

#### 2.1.2. HUVEC Culture

Three different cri-vials containing HUVEC were purchased from PromoCell (Heidelberg, Germany). The provider guaranteed that the cells were tested for cell marker expression of VWF, CD31, and Dil-Ac-LDL uptake assays and they were absent from HIV, HBV, HCV, and microbial contamination. The cryopreserved cells (500,000 cells/cri-vial) immediately upon arrival were plated on 1–10 cm dish in complete EGM2 medium composed of Endothelial cell growth medium 2 (C-22011, PromoCell, Heidelberg, Germany) supplemented with SupplementMix (C-39211, PromoCell, Heidelberg, Germany). Alternatively, the complete EGM2 contains 2% (*v/v*) FBS, 5 ng/mL Epidermal Growth Factor (EGF), 10 ng/mL FGF-2, 20 ng/mL Insulin-like Growth Factor (IGF), 0.5 ng/mL VEGF 165, 1 μg/mL ascorbic acid, 22.5 μg/mL heparin, and 0.2 μg/mL hydrocortisone. Then the cells were expanded in a ratio 1:4 up to P4.

#### 2.1.3. Coculture of BMSC and HUVEC

The direct coculture method at ratio 1:1 was used to cultivate BMSC and HUVEC together. Complete EGM2 medium was utilized to initially coculture both types of cells as EGM2 could increase the plasticity of BMSC [14]. To mimic the inflammatory response, 100 pg/mL IL-1β (R&D Systems, Minneapolis, MN, USA) was added to the cell culture, its concentration used in this study was inferred from literature data [15]. For in vitro hypoxic conditions, either COCl_2_ or DMOG were added to the Complete medium at concentration 100 μM [16] and 0.5 mM [17], respectively. Both COCl_2_ and DMOG were purchased from Sigma (Schnelldorf, Germany). For the differentiation assay, besides COCl_2_ and DMOG, the cells in coculture were also incubated in the hypoxic chamber where oxygen level was maintained at 1% to compare better different hypoxic stimuli.

### 2.2. Cell Proliferation Assay

Cells for monoculture (BMSC only, HUVEC only) and coculture (BMSC + HUVEC) were plated at density 2,000 cells/well on 96 well plates in quintuplicate in different media which are medium Complete and medium Complete supplemented with either 100 pg/mL IL-1β (IL-1β) or 100 μM COCl_2_ (COCl_2_) or 0.5 mM DMOG (DMOG). The media were freshly changed two times per week. At each time point starting from day 1 to day 6, the cells were washed once with 1× PBS, stained with 0.75% g/mL crystal violet solution for 20 min at RT, and finally washed 5 times with distilled water, the powder was supplied by Sigma (Schnelldorf, Germany). Photos of dried cells were taken by using a bright-field microscope (Leica, Wetzlar, Germany), then 100 μL of eluent solution (50% (*v/v*), absolute ethanol, and 1% (*v/v*) acetic acid) was added to each well to dissolve the dye absorbed by the cells, and absorbance was measured at 550 nm by using a microplate reader (Multiskan Ascent, Thermo Labsystems, Midland, Canada).

### 2.3. In Vitro Osteogenic Differentiation

BMSC and HUVEC were cocultured in medium Complete EGM2 at following densities: 50,000 cells/well of 24 well plate (for alizarin Red S staining and immunohistochemistry (IHC)); 300,000 cells/well of 6 well plate for quantitative polymerase chain reaction (qPCR); 700,000 cells/10 cm dish for Enzyme-linked immunosorbent assay (ELISA). When the cells reached total confluence, they were switched to i) osteogenic medium (OM) which is EGM2 containing 2% FBS, 50 μg/mL ascorbic acid (Sigma, Schnelldorf, Germany)), 10mM β glycerolphosphate (Merck, Schnelldorf, Germany)) and 10^−7^ M dexamethasone (Sigma, Schnelldorf, Germany)); ii) OM added 100 100 pg/mL IL-1β; iii) OM supplemented with 100 μM COCl_2,_ iv) OM added 0.5 mM DMOG and v) OM in 1% O_2_. The addition of IL-1β was to evaluate osteogenesis in inflammatory conditions while the supplement of COCl_2_ and DMOG was to examine osteogenic differentiation in hypoxic conditions. Moreover, EGM2 supplemented with 2% FBS only was used as a control (CTR) medium. The media were freshly changed twice per week. The experiment lasted for 9 days.

### 2.4. Alizarin Red S Staining

Alizarin Red S staining was used to observe calcium deposits formed by differentiated cells (osteoblasts) in osteogenic conditions. Cells were washed once with PBS, fixed in 10% formalin for 10 min at RT, washed twice with PBS, subsequently stained in 2% (*w/v*) Alizarin Red S solution pH 4.2 (the powder was supplied by Sigma, Schnelldorf, Germany) for 10 min at RT, and finally washed with distilled water. Photos of the plates containing dry stained cells were taken by using the scanner Epson V200 (Epson, Suwa, Japan).

### 2.5. IHC

The confluent cocultured cells were induced for osteogenesis in four different media as mentioned above. At the end of each time point (day 2 and day 9), cells were washed once with PBS, fixed in 10% formalin for 10 min at RT, then washed again with PBS. 0.2% Triton (Sigma, Schnelldorf, Germany) in PBS was added to the cells for 10 min to permeabilize the membrane. Subsequently, cells were incubated with 4% H_2_O_2_ for 30 min to block the endogenous peroxidase, then in 1 mg/mL hyaluronidase type II (Sigma, Schnelldorf, Germany) in PBS pH 6 for 15 min at 37 °C to loosen the matrix. Then, nonspecific blocking was performed by incubating the cells in pure FBS for 30 min at RT. Immediately after this blocking step, cells were incubated with 1st antibodies against VEGF (1:1000, ab46154, Abcam, Cambridge, UK), VWF (1:500, #65707, Cell Signaling, Beverly, MA, USA), and Collagen type I (1:1500, ab138492, Abcam, Cambridge, UK) for 2 h, RT. The cells were then washed with PBS for 3 × 5 min and incubated with biotinylated anti-rabbit secondary antibody (1:500, Dako Agilent, Santa Clara, CA, USA) for 30 min at RT. The 1st and 2nd antibodies were diluted in 10% FBS in PBS. Negative control (IgG control) sample was also performed by leaving the section in 10% FBS without 1st antibody. After being washed with PBS for 3 × 5 min, the cells were incubated with streptavidin horseradish peroxidase (HRP) conjugate (Dako Agilent, Santa Clara, CA, USA) for 30 min at RT. Subsequently, the cells were washed with PBS for 3 × 5 min. For VEGF and VWF recognition, cells were stained with 3,3′Diaminobenzidine (DAB, Dako Agilent, Santa Clara, CA, USA) for 3 min and counterstained in Mayer’s hematoxylin (Sigma, Schnelldorf, Germany) for 2 min. For Collagen type I detection, the cells were stained with DAB-Nikel (Vector Laboratories, Burlingame, California, CA, USA) without the counterstaining step. Finally, the sections were mounted with DPX mounting medium.

### 2.6. ELISA

ELISA was used to quantify VEGF and VWF released to the media. The confluent cocultured cells were washed twice with sterile PBS, then the dishes were divided into three groups:i)Complete EGM2, Complete EGM2 + COCl_2_, Complete EGM2 + DMOG, Complete EGM2 in 1% O_2_ii)CTR, CTR + COCl_2_, CTR + DMOG, CTR in 1% O_2_iii)OM, OM + COCl_2_, OM + DMOG, OM in 1% O_2_

The cells were incubated in the above media for 24 h, then were washed twice with sterile PBS and maintained in the medium EGM2 containing 0.1% FBS only (2 mL/10 cm dish) for another 24 h. Consequently, the supernatant was collected, centrifuged for 10 min at 12,000 × *g* at 4 °C to remove cell debris and stored at –20 °C. Quantification of VEGF and VWF were measured by using Human VEGF Quantikine ELISA kit (DVE00, R&D Systems, Minneapolis, MN, USA) and Human VWF kit (RAB0556, Sigma, Schnelldorf, Germany), respectively. Absorbance was read at wavelength as recommended instruction of the kits by using a microplate reader (Multiskan Ascent, Thermo Labsystems, Midland, Canada). Each sample was performed in duplicate. The experiments were repeated three times on three different samples.

### 2.7. qPCR

qPCR was performed to analyze the gene expression. Total RNA was isolated by using RNeasy Mini Kit (Qiagen, Hilden, Germany). The quality and quantity of RNA were measured by NanoDrop 8000 (Thermo Fisher Scientific, Wilmington, DE, USA). 1 μg of RNA was reversely transcribed following the instruction of ImProm II reverse Transcription System (Promega, Madison, WI, USA). cDNA was amplified by using the PowerUp SYBR master mix (Thermofisher, Bleiswijk, Netherlands) on the 7500 Fast Realtime PCR System (Applied Biosystems, Waltham, MA, USA). The sequences of primers are listed in Table 1. The reaction plates were initially held at 50 °C for 20 s and then 95 °C for 10 min, subsequently, the cycling stage was performed at 95 °C for 15 s and then 60 °C for 1 min, this cycling stage was repeated for 40 cycles, and finally, the reactions were set at for 95 °C for 15 s, followed by 60 °C for 1 min, 95 °C for 30 s, and 60 °C for 15 s for the melting curve. Gene expression was determined according to the 2(-delta delta C(T)) method [18].

### >2.8. Statistic Analysis

All data are presented as means and standard error of the mean (SEM) of three independent experiments (n = 3). Comparisons were performed by one-way ANOVA followed by post-hoc Tukey’s multiple comparisons test (for ELISA analysis) or two-way ANOVA (all other data). Statistical analysis was performed using GraphPad Prism Version 6.0a software with statistical significance set at *p* < 0.05.

## 3. Results

### 3.1. Proliferation of Cells in Direct Co-Culture System

To examine the viability and proliferation of the cells in coculture, we performed a crystal violet assay. The colorant stains nuclei, quantification of DNA by measuring the absorbance of stained cells at a specific wavelength can infer the cell number. Changes in cell morphology were well observed. Figure 1A shows that before the coculture experiment in the cell expansion phase, BMSC possesses a fibroblast-like shape whereas HUVEC have cobblestone morphologies. Figure 1B illustrates that BMSC and HUVEC in monoculture in media Complete EGM2 and Complete EGM2 supplemented with IL-1β (IL-1β) still maintain their original morphology. In coculture in media Complete EGM2 and IL-1β, there are more rounded cells than elongated ones in spite of the equal ratio of plating cells, this finding correlates with higher growth of HUVEC compared to BMSC resulting in an increased proliferation of cocultured cells in respect of BMSC alone (two left charts in Figure 1C). BMSC in hypoxia-induced by DMOG appear more circular and look healthier than HUVEC alone and cocultured cells under the effect of DMOG, which was confirmed by the far right chart in Figure 1C where the proliferation curve of BMSC reached a peak at the final time point (Day 6). Conversely, BMSC, when cultured alone in the condition containing COCl_2_, suffer from necrosis shown by cytoskeletal disruption, cell swelling and membrane rupture (upper second photo from the right of Figure 1B). However, HUVEC alone under the effect of COCl_2_ absorbed more the violet colorant than those in DMOG or BMSC alone in COCl_2._
Figure 1D shows the relative proliferation of cocultured cells in medium Complete EGM2 or Complete EGM2 supplemented with IL-1β or COCl_2_ or DMOG over time compared to day 1. While the cells in hypoxia induced by chemical agents remained almost at constant growth, the cells in normoxia showed an increased proliferation although there is no significant difference between cells in media Complete EGM2 and IL-1β.

### 3.2. In Vitro Osteogenic Differentiation

First, we evaluated the mineralization of cells in OM conditions in the presence or absence of IL-1β or COCl_2_ or DMOG or in 1% O_2_ over time. Figure 2A shows the result of Alizarin Red S coloration demonstrating that while the cells in conditions OM formed calcium deposit starting from day 7 and visible at day 9, the cells in OM condition with COCl_2_ or DMOG or in 1% O_2_ did not create the deposit even at day 9. Deposit formation was also observed in the cells cultured in the presence of IL-1β. These findings were confirmed by the qPCR result in Figure 2B showing that at the early time of differentiation (day 2), the cells in conditions OM expressed a significantly higher level of ALP gene than those in OM with either COCl_2_ or DMOG or in 1% O_2_. We did not observe the significant change in gene expression of Runx2 (Runt-related transcription factor 2)—a key transcription factor in osteoblast differentiation (data not shown). There was also no significant difference in ALP expression between the cells in OM and IL-1β (data not shown).

While minerals provide stiffness for bone, collagen fibers mostly collagen type I make bone tough and strong [19], thus we believe that evaluation of collagen type I expression is indispensable to analyze osteogenesis. In this experiment, we used qPCR and IHC method to detect collagen type I. While Figure 2B presents that gene expression of collagen type I in the cells in 1% O_2_ is significantly higher compared to that in other conditions, IHC analysis was used to observe the matrix organization and localization of the protein. Figure 2C shows that in monoculture, only BMSC expressed collagen type I (black color). Figure 2D illustrates that the cells incubated in different conditions over time expressed collagen type I but in a different manner. Addition of COCl_2_ or DMOG to the medium and osteogenic differentiation in 1% O_2_ caused the formation of a very loose cellular matrix, conversely, the cells absent from these two reagents created a much denser matrix with strong expression of collagen type I especially by the cells induced in conditions OM and OM supplemented with IL-1β. Moreover, Figure 2D also indicates a change in cell morphology. Cells in the presence of DMOG and 1% O_2_ seem to maintain a rounded shape even at day 9 of induction while the presence of COCl_2_ and oxygen drastically modified cells to become more elongated and smaller. Specifically, the cells in the conditions OM at day 9 differentiated and formed an interlaced net of matrix inferring cross-links of collagen fibers whereas the cell layer in conditions lacking oxygen displayed collagen deterioration.

### 3.3. Expression of Endothelial Cell Marker and Angiogenic Factor in Osteogenic Condition

Besides osteogenesis, we aimed to also evaluate the expression of VWF—one of the endothelial cell markers—and VEGF—a typical marker of angiogenesis of BMSC and HUVEC in 1:1 direct coculture. Figure 3A shows that in monoculture in different conditions, only HUVEC are positive to VWF while Figure 3B demonstrates that in coculture at day 2 VWF was expressed in all conditions even in media CTR and OM supplemented with IL-1β, thus it might say that HUVEC contributes to endothelial marker expression of cocultured cells.

In coculture, at day 2, there was no remarkable difference in VWF appearance, however, at day 9, the cells cultured under the effect of DMOG prominently depicted its presence compared to other conditions. qPCR was also performed to verify the gene expression of this marker by cocultured cells. Figure 3C shows that at day 9 the cells incubated in the presence of DMOG synthesized the highest level of VWF gene that correlates with IHC result, especially the difference was significant compared to conditions OM, OM supplemented with COCl_2_ and OM in 1% O_2_. More importantly, we observed that in monoculture under the effect of hypoxia caused by chemical reagents or at 1% O_2_ at day 9, both BMSC and HUVEC suffered; however, while BMSC were still able to adhere to plastic support, HUVEC detached (data not shown). This observation associated with the IHC result of cocultured cells at day 9 shown in Figure 3B and Figure 4B may suggest that BMSC supports cell attachment in direct coculture. Furthermore, we performed ELISA assay to quantify VWF protein released to supernatant by cocultured cells after a short time (24 h) treatment. In this experiment, besides using OM condition, we also utilized medium Complete EGM2 and CTR added COCl_2_ or DMOG to exclude the possible effect of factors other than hypoxia-induced chemicals COCl_2_ and DMOG. Figure 3D indicates that the concentration of VWF released by the cells cultured in different conditions was similar. There was no significant change in VWF expression analyzed by qPCR and ELISA between the cells cocultured in the presence or absence of IL-1β (data not shown).

Next, we investigated the expression of VEGF. Figure 4A illustrates that both BMSC and HUVEC in monoculture in different conditions over time lightly expressed VEGF. However, in coculture, VEGF was present showing a synergic interaction of the cocultured cells (Figure 4B). Interestingly, at day 2 the cells cultivated with COCl_2_ and DMOG strongly expressed VEGF protein, which correlates with PCR analysis at day 2 in Figure 4C showing an increase of the VEGF gene in the cells stimulated by hypoxia-induced chemicals compared to others. However, at day 9, VEGF production both in protein and gene level of the cells in hypoxia decreased, this contrasts with the cells maintained in normoxia. IHC analysis in Figure 4B indicates that VEGF expression in the cells cocultured in 1% O_2_ at day 9 is less than that in other conditions that contrast with the qPCR result at the same time in Figure 4C. We also performed qPCR to evaluate gene expression of HIF-1α. A lower chart in Figure 4C depicts that after 9 days in coculture the cells in 1% O_2_ expressed the highest level of HIF-1 α gene followed by the cells in DMOG.

In addition, to fully evaluate the expression of VEGF, we also performed the ELISA method to quantify VEGF released into the culture supernatant. Figure 4D shows that both COCl_2_ and DMOG augmented VEGF release but DMOG has a much more striking impact. In all three groups of treatment conditions (Complete EGM2, CTR, and OM), DMOG led to a significant increase of VEGF compared to 1% O_2_ and COCl_2_ (*p* < 0.05). This finding could imply that in terms of using hypoxia-induced chemicals to increase angiogenesis, DMOG could be a better option than COCl_2_. We also experimented with using IL-1β to observe the expression of VEGF, like results in osteogenesis and VWF expression, the addition of IL-1 β does not impede VEGF expression (data not shown).

## 4. Discussion

Bone fracture healing of long bones is a complex and dynamic process orchestrated by three main phases namely inflammation, reparation and remodeling [20,21]. In inflammatory phase, blood vessels carry cells of the immune system including platelets, neutrophils, and macrophages to the wound site, in turn, these cells are activated to release inflammatory cytokines to attract other cells. In the reparative step, the periosteal cells migrate and differentiate into chondrocytes at the wound to make callus followed by the matrix calcification, angiogenesis and substitution of chondrocytes by osteoblast. Finally, remodeling occurs to form compact bone and maturation of osteoblasts to osteoclast. Thus, the blood vessel is considered as the key element to initialize fracture healing and plays a critical role in cell migration, proliferation, and early differentiation.

In total hip arthroplasty, a metal prosthesis connects the proximal femur with the acetabulum to replace the physiological hip function. However, over time a revision surgery may be required mainly due to prostheses instability and mechanical loosening [22]. In the revision procedure, synthetic bone grafts could be used to increase bone stock to favor integration of the host body and prostheses [23]. Recently, bone tissue engineering has received much interest as it uses synthetic bone and cells to induce osteogenesis and osteoinduction [24].

To increase the number of mammalian cells in vitro, scientists often incubate them in normoxia, which provides atmospheric oxygen level (20%–21%), however physiological oxygen in tissues falls in 2%–9% that can be considered as hypoxia, particularly acute hypoxia (1% O_2_) can exist at some tissues such as kidney, bone marrow, thymus [4]. It should be noted that the environment in the bone fracture site is hypoxic because of blood extravasation leading to interruption of oxygen supply and strong consummation of oxygen of cells recruited to the wound [25], thus the role of oxygen in bone healing should be considered. In this study, we used CoCl_2_ and DMOG to stimulate hypoxia. CoCl_2_ inhibits von Hippel–Lindau protein (*p*VHL) binding to oxygen degrading domain consisting of proline residues of HIF-1α thus preventing HIF-1α hydroxylation by oxygen-dependent prolyl hydroxylases (PHD) [1,26] whereas DMOG directly inhibits PHD leading to HIF-1α stability [27,28].

We investigated the effect of hypoxia on cell proliferation, osteogenesis and angiogenesis in vitro of direct coculture system of BMSC—the progenitor cells able to differentiate into osteoblast and chondrocyte—and HUVEC—endothelial cells widely used for the study of blood vessel formation. Our findings are summarized as follows:

1. We observed that the presence of atmospheric oxygen can promote viability and proliferation of cells in both monoculture and coculture, thus for expansion of BMSC and HUVEC together in normoxia is superior to in hypoxia. Endothelial cells in the endothelium are the first cells in contact with blood, thus changes in oxygen concentration of the blood could affect the cell types. Our evaluation of the impact of oxygen level on human BMSC proliferation is consistent with findings of Holzwarth et al., 2010 [17] but in contrast with Grayson et al., 2007 [29]. In this study, we used the complete medium containing growth factor FGF-2 for BMSC culture, Holzwarth et al., 2010 [17] used 5% (*v/v*) human plasma that also consists of a cocktail of growth factors and cytokines whereas Grayson et al., 2007 [29] did not use any supplement of growth factors except for standard FBS. We hypothesize that the effect of oxygen on human BMSC proliferation is dependent on components in the cell culture medium. BMSC slowly grows in culture, the addition of factors required for proliferation provided that they maintain cell stemness should be recommended.

2. Normoxia benefits in vitro bone formation of BMSC and HUVEC in coculture while hypoxia promotes angiogenesis. BMSC and HUVEC were cocultured up to full confluence then induced to osteogenic differentiation. Calcium deposits produced by osteoblast during mineralization are an indication of successful in vitro osteogenesis. In normoxia, after 9 days, the cells formed nice calcium deposits showed by red color after staining with Alizarin Red S whereas the cells in hypoxia did not. Expression of genes specific for bone formation (ALP) was also higher in normoxia than in hypoxia. Moreover, in both normoxia and hypoxia, we observed the appearance of collagen type I—marker of extracellular matrix of bone, but the matrix of the cells in normoxia was much denser particularly after 9 days in culture showing strong crosslinking of proteins and matrix maturation. Considering that collagen type I is the most abundant protein accounting for more than 90% of bone mass and provides tensile strength as well as load-bearing for bone connective tissue [30] thus it is indispensable to evaluate its expression in the study of bone formation.

Simultaneously with osteogenesis examination, we also investigated VWF expression of cells in osteogenic medium. All of the cells expressed this endothelial marker, however over time the cells stimulated in hypoxia indicated stronger signal. In particular, DMOG could be more efficient compared to CoCl_2_ as the cells incubated in DMOG produced more intense brown color presenting for VWF than CoCl_2_, this also correlates with the qPCR result. In addition, we discovered that short incubation (24 h) in hypoxia does not change the concentration of VWF released by the cells compared to normoxia.

VEGF is the fundamental element of angiogenesis—new blood vessel formation from existing vasculature [31,32]. All of the cells incubated in both normoxia and hypoxia strongly expressed VEGF especially the cells stimulated by CoCl_2_ and DMOG after 2 days of treatment. More surprisingly, we discovered that after a short treatment (24 h) concentration of VEGF released into the culture supernatant produced by cells incubated in hypoxia surged particularly in the cells stimulated by DMOG, while the level of VWF released by cells remained unchanged between normoxia and hypoxia. Importantly, by profiling VWF and VEGF expression of the cells under the effect of two chemicals simulating hypoxia, we found that DMOG could be a better candidate compared to CoCl_2_. Considering that long hypoxia (more than 3 days) could hinder cell proliferation and viability, short hypoxia (1 day) is sufficient to push VEGF production while not affecting VWF expression and normoxia is critical for osteogenesis, thus to obtain both bone and blood formation, normoxia should be applied initially for cell culture to induce osteogenesis followed by short treatment in hypoxia to facilitate angiogenesis.

3. Local inflammation is the first phase in wound healing, thus the presence of pro-inflammatory cytokines could be beneficial to trigger successive phases in the healing process. Thus, we used IL-1β to mimic the inflammatory environment. We found that the addition of IL-1 β to the medium at 100 pg/mL concentration increased cell proliferation, did not interfere with osteogenic differentiation and did not alter angiogenesis of BMSC and HUVEC cocultured in normoxia. However, the level of cytokines must be limited otherwise systematic inflammation could occur and cause destruction of tissue matrix because of the production of matrix metalloproteinases (MMPs) [33,34]. This observation may imply that implantation of scaffolds seeded with BMSC and HUVEC in the inflamed joint could be possible, however further studies on other cytokines and longer time duration (more than 9 days) should be done.

BMSC represents a possible application for patients suffering from musculoskeletal injuries. Our preliminary study showed that direct coculture of BMSC and endothelial cells at ratio 1:1 could maintain both osteogenesis and angiogenic potentials depending on oxygen concentration in culture, which could be modified by the addition of hypoxia-induced reagents. Further studies both in vitro and in vivo are needed to better understand the complex relationship between osteogenesis and angiogenesis to generate newly engineered scaffolds able to promote bone regeneration.

## Figures and Tables

**Figure 1 cells-09-00757-f001:**
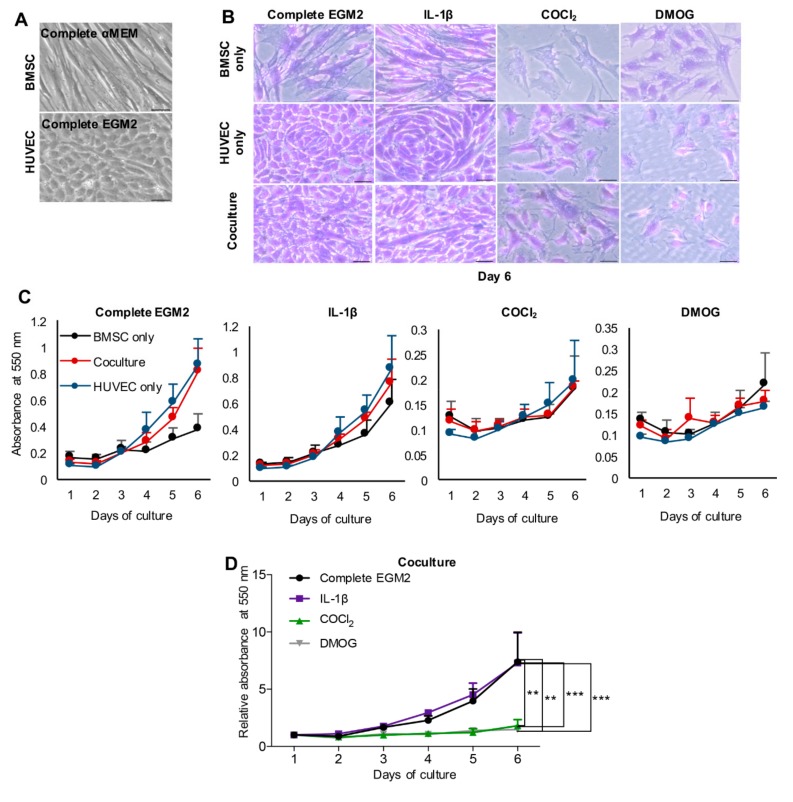
Cell proliferation. (**A**) Morphology of bone marrow-derived mesenchymal stem cells (BMSC) and human umbilical vein endothelial cells (HUVEC) cultured in media Complete αMEM and Complete EGM2, respectively, for cell expansion. (**B**) Morphology of cells in monoculture and coculture in Complete EGM2 medium in the presence/absence of IL-1β or COCl_2_ or DMOG, scale bar 50 μm. (**C**) Proliferation curve of the cells cultured in Complete EGM2 medium supplemented with IL-1β or COCl_2_ or DMOG determined by measuring the absorbance of the cells stained with crystal violet at 550 nm wavelength. (**D**) Relative proliferation of cocultured cells in different media over time compared to day 1. Results are the average + SEM of three experiments performed in quintuplicate from three different cell cultures. Lower chart demonstrates relative proliferation of cocultured cells over time normalized to day 1 (∗∗ *p* ≤ 0.0018, ∗∗∗ *p* ≤ 0.0005).

**Figure 2 cells-09-00757-f002:**
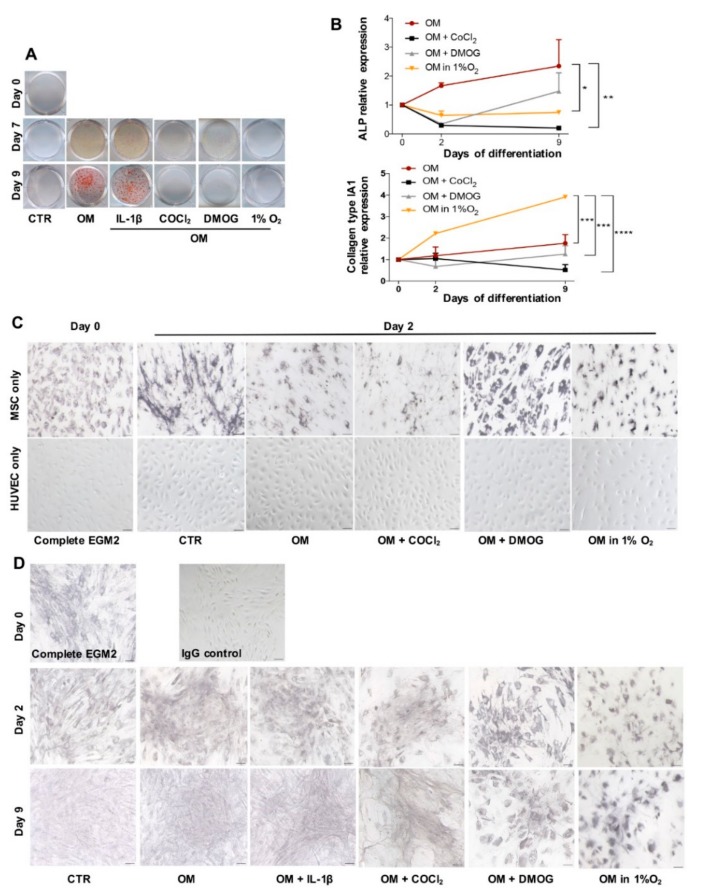
In vitro osteogenesis. (**A**) Alizarin Red S coloration shows the accumulation of calcium deposit (red) of cells cocultured in control medium (CTR), osteogenic medium (OM), OM added IL-1β or COCl_2_ or DMOG and OM in 1% O_2_ at different time points (day 0, 7, 9). (**B**) qPCR analysis of gene expression of ALP and collagen type I over time (day 0, 2, 9) in conditions OM, OM added COCl_2_ or DMOG and OM in 1% O_2_ (n = 3, ∗ *p* = 0.0156, ∗∗ *p* < 0.0025, ∗∗∗ *p* = 0.0002, ∗∗∗∗ *p* < 0.0001). (**C**) IHC analysis illustrates Collagen type I expression (black) of BMSC and HUVEC in monoculture in medium Complete EGM2 at day 0 and in day 2 in media CTR, OM, OM supplemented with COCl_2_ or DMOG and OM in 1% O_2_. Counterstaining was skipped, scale bar 50 μm. (**D**) IHC analysis of cocultured cells over time in different conditions. IgG control means the section was not incubated with 1st antibody, scale bar 50 μm.

**Figure 3 cells-09-00757-f003:**
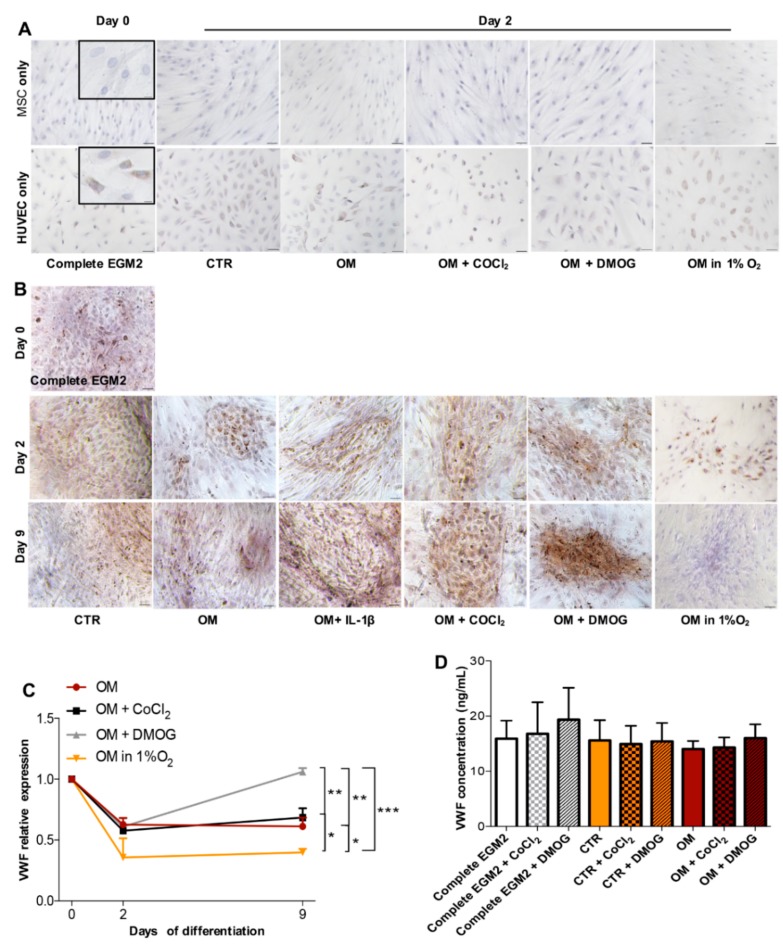
Expression of endothelial cell marker—VWF. (**A**) IHC analysis of VWF expressed by BMSC and HUVEC in monoculture at day 0 in medium Complete EGM2 and at day 2 in media CTR, OM, OM supplemented with COCl_2_ or DMOG and OM in 1% O_2,_ scale bar 50 μm, inset scale bar 25 μ m. (**B**) IHC analysis of VWF expression by cocultured cells in different conditions at day 2 and day 9, scale bar 50 μ m. (**C**) Relative gene expression of VWF shown by qPCR (n = 3, ∗ *p* ≤ 0.0228, ∗∗ *p* ≤ 0.0057, ∗∗∗ *p* = 0.0006). (**D**) Quantification of VWF released to the medium performed by the ELISA method. The cells were incubated in media Complete EGM2, CTR and OM in the presence/absence of COCl_2_ or DMOG for 24 h followed by another 24 h in the medium containing 0.1% FBS only (n = 3, *p* > 0.05).

**Figure 4 cells-09-00757-f004:**
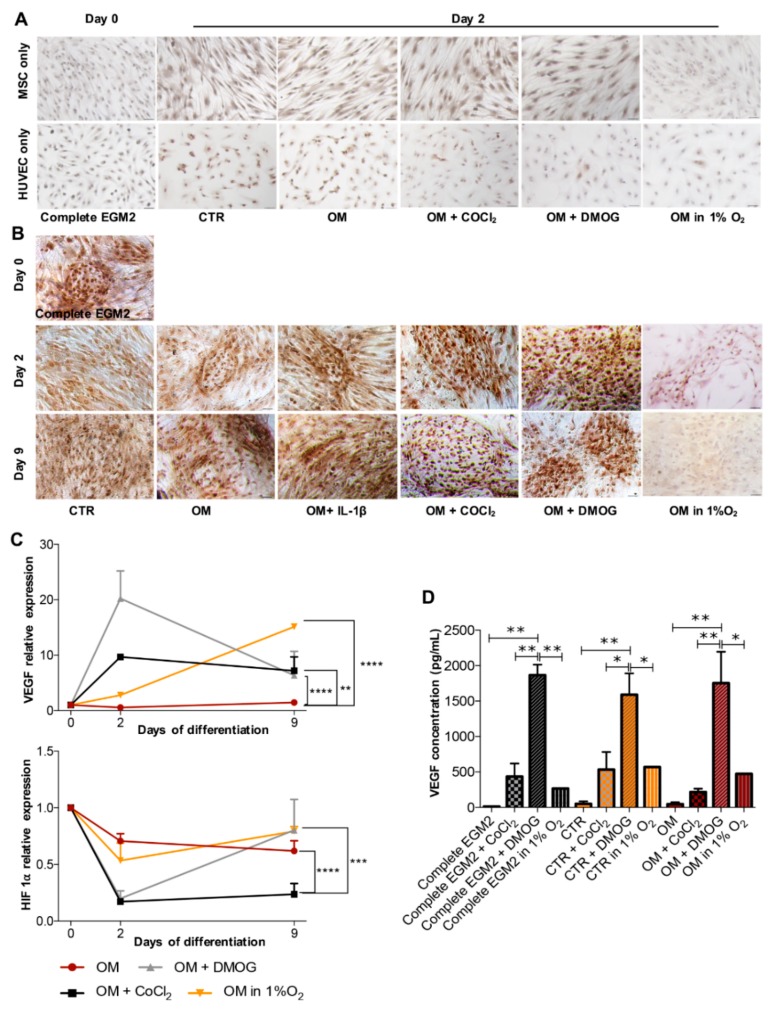
In vitro angiogenesis. (**A**) IHC analysis of vascular endothelial growth factor (VEGF) (brown) expressed by BMSC and HUVEC in monoculture at day 0 in medium Complete EGM2 and at day 2 in media CTR, OM, OM supplemented with COCl_2_ or DMOG and OM in 1% O_2_, scale bar 50 μ m. (**B**) IHC result of VEGF expression by the cocultured cells, scale bar 50 μm. (**C**) qPCR presents relative gene expression of VEGF and HIF 1α of the cells cocultured in media OM, OM supplemented with COCl_2_ or DMOG and OM in 1% O_2_ over time (n = 3, ∗∗ *p* ≤ 0.0029, ∗∗∗ *p* = 0.0004, ∗∗∗∗ *p* < 0.0001). (**D**) ELISA data shows the concentration of VEGF released in the supernatant by the cocultured cells in three groups: Complete EGM2, CTR, and OM. In each group, the cells were incubated with or without COCl_2_ or DMOG and in 1% O_2_ for 24 h then in the medium supplemented with 0.1% FBS only for another 24 h (n = 3, ∗ *p* < 0.05, ∗∗ *p* < 0.01).

**Table 1 cells-09-00757-t001:** Primer sequences for qPCR (F/R: forward/reverse).

Human Gene	Sequence (5′-3′)	Product Size (bp)
18S ribosomal RNA (18S rRNA)	F: GTAACCCGTTGAACCCCATT	151
	R: CCATCCAATCGGTAGTAGCG	
Collagen type IA1	F: CAGCCGCTTCACCTACAGC	83
	R: TTTTGTATTCAATCACTGTCTTGCC	
Alkaline phosphatase (ALP)	F: ACCACCACGAGAGTGAACCA	79
	R: CGTTGTCTGAGTACCAGTCCC	
HIF-1α	F: CATAAAGTCTGCAACATGGAAGGT	148
	R: ATTTGATGGGTGAGGAATGGGTT	
VEGF	F: CTACCTCCACCATGCCAAGT	109
	R: GCAGTAGCTGCGCTGATAGA	
VWF	F: GCTGCTGGACACAAGTTTGA	237
	R: ACTCATGGGGCTCTGCATAC	

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
