# Peer review of "Effect of Chemically Induced Hypoxia on Osteogenic and Angiogenic Differentiation of Bone Marrow Mesenchymal Stem Cells and Human Umbilical Vein Endothelial Cells in Direct Coculture"

_cells, 2020, doi:10.3390/cells9030757_

Round 1

Reviewer 1 Report

The combining of tissues that primarily have different local effects on the environment of bone formation is very innovative. What is unclear is the hypoxia and why two different types of hypoxia are used- environmental and chemical. This is not well described in the manuscript.

Discussion point 3, the  potential of affecting osteoarthritis with these different types of cells is not related to the topic of bone regeneration. In fact angigenesis should not play a role in hyaline cartilage regeneration since it is an avascular tissue with the likelihood of calcification if invaded by blood vessels, as in the growth plate. My suggestion is not to discuss osteoarthritis in this manuscript.

Author Response

Reviewer 1:

The combining of tissues that primarily have different local effects on the environment of bone formation is very innovative. What is unclear is the hypoxia and why two different types of hypoxia are used- environmental and chemical. This is not well described in the manuscript.

  • Answer 1 to the comment “What is unclear is the hypoxia”

In the 1st submitted manuscript, lines 396-401, we wrote that “physiological oxygen in tissues falls in 2-9% that can be considered as hypoxia, particularly acute hypoxia (1% O2) can exist at some tissues such as kidney, bone marrow, thymus [4]. It should be noted that the environment in the bone fracture site is hypoxic because of blood extravasation leading to interruption of oxygen supply and strong consumption of oxygen of cells recruited to the wound [33], thus role of oxygen in bone healing should be considered.” The condition in bone marrow and at the site of bone fracture is hypoxic according to literature. In our study, MSC were isolated from bone marrow from femoral canal of patients undergoing hip replacement procedure. The operational site of the patient can be considered as a bone fracture site since femoral head of femur is removed. We wanted to mimic the hypoxic condition of bone marrow and bone fracture site, thus we cultured cells in normoxia and chemical induced hypoxia.

  • Answer 2 to the commentwhy two different types of hypoxia are used- environmental and chemical

In the introduction section in the 1st submitted manuscript, lines 68-74, we mentioned disadvantages of hypoxic incubator/chamber that have been used to study hypoxia. “It should be noted that hypoxic incubator cannot guarantee complete hypoxia since once the incubator’s door is open, the cells inside could be in contact with oxygen and it also requires time for the incubator to lower the oxygen back to needed level. Although hypoxic chamber overcomes this drawback since working bench and trays for placing cell plates are inside hypoxic chamber, it can be limited solely for pilot experiment. To address these problems, we used two reagents previously known for simulating hypoxia namely Cobalt(II) chloride (CoCl2) and Dimethyloxaloylglycine (DMOG) for coculture system to compare with in normoxia”. Thus we proposed the use of chemical agents that stimulate hypoxia. Moreover, in the literature other scientists used hypoxic incubator/chamber to study cells in coculture, thus we did not want to repeat already published. Therefore, we only used hypoxic chamber as a control for the use of chemical agents.

Discussion point 3, the potential of affecting osteoarthritis with these different types of cells is not related to the topic of bone regeneration. In fact angigenesis should not play a role in hyaline cartilage regeneration since it is an avascular tissue with the likelihood of calcification if invaded by blood vessels, as in the growth plate. My suggestion is not to discuss osteoarthritis in this manuscript.

  • Answer 3

We thank the reviewer for the suggestion. We changed discussion point 3 accordingly.

Reviewer 2 Report

The manuscript entitled „Effect of chemically induced hypoxia on osteogenic and angiogenic differentiation of bone marrow mesenchymal stem cells and human umbilical vein endothelial cells in direct coculture” by Thi Nguyen et al is of interest for the readers of cells.

In the present manuscript, the effect of hypoxia (chemically induced and by low oxygen) on the osteogenic properties of MSC, angiogenic characteristics of HUVEC in single and coculture was assessed over period of up to 10 days. It was observed that chemically induced hypoxia decreased proliferation and survival and affect osteogenic differentiation negatively whereas angiogenic properties were raised. Coculture conditions did not improve osteogenesis in hypoxic conditions but angiogenic characteristics were not affected.

An original manuscript is provided, as proved by pubmed and google scholar using the search terms “hypoxia, MSC, HUVEC, coculture, Cobalt(II)chloride, Dimethyloxaloglycine” in different combinations. In general, a classical cell biological study is presented. The aim of the study is relevant, since cell therapeutic approaches in the bone healing often suffer from massive initial hypoxia and there is a high demand to address this topic. Furthermore, ‘real’ hypoxia, induced by low oxygen, is compared to two approaches of chemically induced hypoxia which is also important for the research in this field. The experiments are generally comprehensive, and well described, however, some questions are still open, as will be shown below.

  1. The wnt/b-catenin pathway is analysed. In my opinion this analysis and its way of presentation is hampered by some shortcomings. The authors presented a western blot depicting b-catenin in nuclear proteins. No densitometry is presented. Furthermore it would be eligible to analyse expression of genes controlled by b-catenin or to inhibit b-catenin by compounds and analyse the effect on angiogenic and osteogenic properties during hypoxia. Actually, it is not possible to conclude anything regarding the role of b-catenin.
  2. Introduction, line 68-77: I think the information about the mode of action of the hypoxic substances could be better placed into the Discussion section.
  3. Materials and Methods: 117-126. It is not entirely clear which medium was used for MSC in the experiments. In co-culture endothelial cell medium was used. But which medium was used when MSC were cultured alone (a-MEM as described in line 99)?
  4. What is the influence of the endothelial medium on the differentiation of MSC? It is described that MSC can acquire endothelial characteristics when cultured under angiogenic conditions. If this is the case, how does this affect osteogenic differentiation?
  5. line 139: Osteogenic differentiation. Usually time span for osteogenic differentiation and detection of calcium deposits is 14 to 21 days. Why did you choose nine days?
  6. line 230: Statistic analysis. Unpaired t-test was performed. But at least four groups were compared simultaneously (e.g. Figure 1B) which demands a test for multiple comparisons followed by a posthoc analysis. Statistic should be recalculated and revised accordingly.
  7. Results: Figure 1A: I miss images (and description) of the cell’s phenotype, proliferation, b-catenin-expression cultured in 1% oxygen (for all other analyses presented in figures 2, 3, 4 this condition is shown).
  8. Results: Collagen-expression, line 268 -301: I wonder why col1A was not additionally analysed by qPCR which can be better quantified than the col1A deposition. If possible, please provide additional COL1A gene expression data.
  9. Results: vWF-expression, Figure 3b: Is it possible to distinguish between HUVEC and MSC in coculture to determine, if MSC also express vWF as it is suggested by the authors? Can this be demonstrated in images made with higher magnification?
  10. Results: VEGF-expression, Figure 4: See comment 9.
  11. Line 359 (and also frequently in the Discussion): Do not use the term “circulating” for factors released into the culture supernatant. They were presumably not circulating.
  12. Discussion (line 464-468): This sentence is hard to understand. Please reword or split into two sentences.

Author Response

Reviewer 2:

  1. The wnt/b-catenin pathway is analysed. In my opinion this analysis and its way of presentation is hampered by some shortcomings. The authors presented a western blot depicting b-catenin in nuclear proteins. No densitometry is presented. Furthermore it would be eligible to analyse expression of genes controlled by b-catenin or to inhibit b-catenin by compounds and analyse the effect on angiogenic and osteogenic properties during hypoxia. Actually, it is not possible to conclude anything regarding the role of b-catenin.
  • Answer 1: Thank you for your comment. As you explained, we think that the addition b-catenin to this study is not necessary, we would like to perform another separate study to use compounds inhibiting its expression. We deleted the parts related to b-catenin in the revised manuscript.
  1. Introduction, line 68-77: I think the information about the mode of action of the hypoxic substances could be better placed into the Discussion section.
  • Answer 2: We moved the part of the action mode of hypoxic substances in the Discussion section as you suggested. We would like to maintain the lines 68-74 in the Introduction section that describe the disadvantages of hypoxic incubator/chamber that have led us propose of using hypoxic agents.
  1. Materials and Methods: 117-126. It is not entirely clear which medium was used for MSC in the experiments. In co-culture endothelial cell medium was used. But which medium was used when MSC were cultured alone (a-MEM as described in line 99)?
  • Answer 3: We used a-MEM only for initial expansion of MSC (section 2.1.1). In co-culture endothelial cell medium (EGM2) was used. To evaluate the synergic effect of HUVEC and MSC in co-culture, we also performed monoculture in EGM2. We corrected legends of the figures to make it clearer.
  1. What is the influence of the endothelial medium on the differentiation of MSC? It is described that MSC can acquire endothelial characteristics when cultured under angiogenic conditions. If this is the case, how does this affect osteogenic differentiation?
  • Answer 4: In the article “Pericyte-Like Progenitors Show High Immaturity and Engraftment Potential as Compared with Mesenchymal Stem Cells”, Amina Bouacida and colleagues used endothelial medium-EGM2-to culture MSC and they demonstrated that “endothelial medium showed similar characteristics (MSC markers and adipo-osteo-chondroblastic differentiation potential)” like mesenchymal stem cell medium. For this reason, we only used a-MEM for initial expansion of MSC based on findings of our previous article and we think that a-MEM has been widely used for MSC expansion. However, for co-culture experiment, we used EGM2 based on results of Amina Bouacida’s article. In fact, Alizarin Red S analysis in figure 2.A of our manuscript showed that the cells cocultured in EGM2 then switched in osteogenic medium (OM) were able to form calcium deposit – an indicator of osteogenesis.
  1. line 139: Osteogenic differentiation. Usually time span for osteogenic differentiation and detection of calcium deposits is 14 to 21 days. Why did you choose nine days?
  • Answer 5: In the 1st submitted manuscript, lines 335-337, we wrote that “More importantly, we observed that in monoculture under the effect of hypoxia caused by chemical reagents or at 1% O2 at day 9, both BMSC and HUVEC suffered; however, while BMSC were still able to adhere to plastic support, HUVEC detached (data not shown)”. We would like to compare osteogenesis of MSC and HUVEC in coculture in normoxia and hypoxia. Unfortunately, long exposure to hypoxia is toxic for the cells, so they died and detached, therefore we chose 9 days instead of 14 to 21 days as usual.
  1. line 230: Statistic analysis. Unpaired t-test was performed. But at least four groups were compared simultaneously (e.g. Figure 1B) which demands a test for multiple comparisons followed by a posthoc analysis. Statistic should be recalculated and revised accordingly.
  • Answer 6: Thank you for your comment. We recalculated by using one-way ANOVA followed by post-hoc Tukey’s multiple comparisons test (for ELISA analysis) or two-way ANOVA (all other data). Please see section 2.8 and charts of figures
  1. Results: Figure 1A: I miss images (and description) of the cell’s phenotype, proliferation, b-catenin-expression cultured in 1% oxygen (for all other analyses presented in figures 2, 3, 4 this condition is shown).
  • Answer 7: Our title is “Effect of Chemically Induced Hypoxia on Osteogenic and Angiogenic Differentiation of Bone Marrow Mesenchymal Stem Cells and Human Umbilical Vein Endothelial Cells in Direct Coculture”. We focused on osteogenic and angiogenic differentiation of cocultured cells in hypoxia stimulated by chemical agents compared with those in normoxia. We did not name the title related to general hypoxia that includes 1% oxygen. However, with focus on osteogenic and angiogenic differentiation, we believed that it is necessary to perform experiments of cell differentiation in 1% oxygen as a control and we did it. Regarding cell proliferation, as answered to your comment No. 5, we mentioned cell toxicity in hypoxia including 1% oxygen, we presumed that 1% oxygen has similar effect on cell proliferation like chemically induced hypoxia, thus we did not perform this experiment.
  1. Results: Collagen-expression, line 268 -301: I wonder why col1A was not additionally analysed by qPCR which can be better quantified than the col1A deposition. If possible, please provide additional COL1A gene expression data.
  • Answer 8: We performed qPCR for COL1A as you suggested, please see Figure 2 and section 3.2 in the revised manuscript.
  1. Results: vWF-expression, Figure 3b: Is it possible to distinguish between HUVEC and MSC in coculture to determine, if MSC also express vWF as it is suggested by the authors? Can this be demonstrated in images made with higher magnification?
  • Answer 9: In the 1st submitted manuscript, lines 315-318, we wrote that “Figure 3A shows that in monoculture in different conditions, only HUVEC are positive to VWF while Figure 3B demonstrates that in coculture at day 2 VWF was expressed in all conditions even in media CTR and OM supplemented with IL-1β, thus it might say that HUVEC contribute to endothelial marker expression of cocultured cells”. We did not suggest that MSC express VWF. We performed IHC analysis of VWF on monoculture as shown in Figure 3A to evaluate VWF expression of each cell type. Figure 3A shows that MSC alone does not express VWF. We provided figure 3 in the revised manuscript with higher resolution.
  1. Results: VEGF-expression, Figure 4: See comment 9.
  • Answer 10: We did not label the cells before coculture. Figure 4A shows that both MSC and HUVEC express VEGF but very light when compared to that in coculture shown in Figure 4B, thus we propose that MSC and HUVEC have synergic effect to increase VEGF expression in coculture. We provided figure 4 in the revised manuscript with higher resolution.
  1. Line 359 (and also frequently in the Discussion): Do not use the term “circulating” for factors released into the culture supernatant. They were presumably not circulating.
  • Answer 11: We did not use the word “circulating” in the revised manuscript as you suggested.
  1. Discussion (line 464-468): This sentence is hard to understand. Please reword or split into two sentences.
  • Answer 12: We split into two sentences which are “Considering that as mentioned above, local inflammation is the first phase in wound healing, thus presence of pro-inflammatory cytokines could be beneficial to trigger successive phases in the healing process. However, the level of cytokines must be limited otherwise systematic inflammation could occur and cause destruction of tissue matrix because of production of matrix metalloproteinases (MMPs)”.

Round 2

Reviewer 2 Report

All my questions and concern were adequately adressed.